# Impact of Dietary Modifications on Plasma Sirtuins 1, 3 and 5 in Older Overweight Individuals Undergoing 12-Weeks of Circuit Training

**DOI:** 10.3390/nu13113824

**Published:** 2021-10-27

**Authors:** Paulina Wasserfurth, Josefine Nebl, Miriam Rebekka Rühling, Hadeel Shammas, Jolanthe Bednarczyk, Karsten Koehler, Tim Konstantin Boßlau, Karsten Krüger, Andreas Hahn, Anibh Martin Das

**Affiliations:** 1Faculty of Natural Sciences, Institute of Food Science and Human Nutrition, Leibniz University Hannover, 30167 Hannover, Germany; paulina.wasserfurt@tum.de (P.W.); nebl@nutrition.uni-hannover.de (J.N.); hahn@nutrition.uni-hannover.de (A.H.); 2Department of Sport and Health Sciences, Technical University of Munich, 80992 Munich, Germany; karsten.koehler@tum.de; 3Clinic for Paediatric Kidney, Liver and Metabolic Diseases, Hannover Medical School, 30625 Hannover, Germany; Miriam.Rebekka.Das@tiho-hannover.de (M.R.R.); shammas.hadeel@mh-hannover.de (H.S.); Bednarczyk.Jolanthe@mh-hannover.de (J.B.); 4Department of Exercise Physiology and Sports Therapy, Institute of Sports Science, Justus-Liebig-University Giessen, 35394 Giessen, Germany; Tim.K.Bosslau@med.uni-giessen.de (T.K.B.); Karsten.Krueger@sport.uni-giessen.de (K.K.)

**Keywords:** aging, sirtuins, combined training, diet, exercise

## Abstract

Sirtuins are nicotinamide adenine dinucleotide (NAD+)-dependent deacetylases that regulate numerous pathways such as mitochondrial energy metabolism in the human body. Lower levels of these enzymes were linked to diseases such as diabetes mellitus and were also described as a result of aging. Sirtuins were previously shown to be under the control of exercise and diet, which are modifiable lifestyle factors. In this study, we analyzed SIRT1, SIRT3 and SIRT5 in blood from a subset of healthy elderly participants who took part in a 12-week randomized, controlled trial during which they performed, twice-weekly, resistance and aerobic training only (EX), the exercise routine combined with dietary counseling in accordance with the guidelines of the German Nutrition Society (EXDC), the exercise routine combined with intake of 2 g/day oil from *Calanus finmarchicus* (EXCO), or received no treatment and served as the control group (CON). In all study groups performing exercise, a significant increase in activities of SIRT1 (EX: +0.15 U/mg (+0.56/−[−0.16]), EXDC: +0.25 U/mg (+0.52/−0.06), EXCO: +0.40 U/mg (+0.88/−[−0.12])) and SIRT3 (EX: +0.80 U/mg (+3.18/−0.05), EXDC: 0.95 U/mg (+3.88/−0.55), EXCO: 1.60 U/mg (+2.85/−0.70)) was detected. Group comparisons revealed that differences in SIRT1 activity in EXCO and EXDC differed significantly from CON (CON vs. EXCO, *p* = 0.003; CON vs. EXDC, *p* = 0.010). For SIRT3, increases in all three intervention groups were significantly different from CON (CON vs. EX, *p* = 0.007; CON vs. EXDC, *p* < 0.001, CON vs. EXCO, *p* = 0.004). In contrast, differences in SIRT5-activities were less pronounced. Altogether, the analyses showed that the activity of SIRT1 and SIRT3 increased in response to the exercise intervention and that this increase may potentially be enhanced by additional dietary modifications.

## 1. Introduction

Aging is associated with changes in human energy and cell metabolism, which may result in metabolic dysfunction (e.g., diabetes mellitus), increased inflammation or the accumulation of senescent cells [1,2]. One modulator of metabolic pathways and cellular stress response are sirtuins.

Sirtuins are a family of nicotinamide adenine dinucleotide (NAD+)-dependent deacetylases of which seven have been identified in mammals (SIRT1–SIRT7). While SIRT1, SIRT6 and SIRT7 are primarily localized within the nucleus, SIRT3–SIRT5 are found in the mitochondria and SIRT2 in the cytosol [1]. At the molecular level, the main function of sirtuins is the modification of proteins at their lysine residues [2]. Though the main enzymatic reaction is (NAD+)-dependent deacetylation, other reactions such as ADP-ribosylation are also catalyzed [3]. SIRT5 was shown to have a weaker deacetylation activity and predominantly functions as desuccinylase, demalonylase and deglutarylase [3,4,5].

Exercise results in an increased energy turnover, especially in skeletal and heart muscle. The energy demand may vary up to 10-fold in the heart [6] and up to 100-fold in skeletal muscle [7], and is mainly met by mitochondrial oxidative phosphorylation. Subsequently, the oxygen consumption varies > 4.5-fold in the heart, up to 17-fold in skeletal muscle, and 25% in the liver as well as the kidneys [8]. Energy supply has to vary commensurate with energy demand; several mechanisms are operative to fine-tune energy metabolism. Passive regulation of the mitochondrial respiratory chain occurs by feed-forward regulation via substrate (adenosine diphosphate [ADP]) saturation with enhanced physical exercise leading to increased ADP (substrate) levels. Furthermore, active regulation of the mitochondrial ATP synthase (complex V) has been demonstrated with mitochondrial electrochemical potential and calcium acting as regulatory elements [9]. Increasing calcium levels during exercise activates enzymes of the citric acid cycle [10,11,12]. In this regard, sirtuins, namely sirtuin 1 (SIRT1) and sirtuin 3 (SIRT3) activity, have been shown to increase during exercise, which results in enhanced energy metabolism (for a recent review see: [13]). Recently, we have observed exercise-induced up-regulation of sirtuins in humans [14].

In addition to exercise, sirtuins are also under the control of diet. The most studied dietary modification linked to an increased life span and upregulation of sirtuins is caloric restriction (CR). Particularly, SIRT1 was extensively studied and shown to respond to CR [15,16,17]. Moreover, other dietary compounds such as polyphenols were also demonstrated to promote SIRT1 activity [18]. Polyphenols are commonly found in fruits and vegetables but also seeds and whole-grains, as well as coffee, green tea or wine [19].

Other dietary compounds which have gained attention are the omega-3 polyunsaturated fatty acids (n3-PUFAs), eicosapentaenoic acid (EPA) and docosahexaenoic acid (DHA). While some studies linked them to improved mitochondrial function [20,21,22], another study showed decreased mitochondrial enzyme activity [23]. Regarding SIRT3, EPA is discussed to enhance SIRT3 expression and therefore positively impact mitochondrial oxidative capacity [24]. Moreover, n3-PUFAs were shown to elicit anti-inflammatory effects through activation of SIRT1 pathways [25,26]. Furthermore, DHA was linked to SIRT1-dependent improvement in endothelial function [27].

One novel source of *n*-3 PUFAs is oil from the copepod *Calanus finmarchicus*, which contains fatty acids mainly bound as wax esters. In addition, this marine oil also contains the antioxidant astaxanthin which may have additional beneficial effects [28].

In a recent study, we could show, that an acute bout of exercise differently affected sirtuin activity in recreational runners consuming an omnivorous, lacto-ovo vegetarian or vegan diet [14]; in the omnivorous participants changes in SIRT1, SIRT3 and SIRT5 were reported.

Based on the existing literature and our previous findings we hypothesized that (1) chronic exercise over 12-weeks would upregulate sirtuins, and that (2) dietary modifications would further enhance exercise-induced upregulation. To test these hypotheses, we analyzed a subset of samples from elderly participants who took part in a 12-week interventional trial. During the study, participants only performed a combined resistance and aerobic training twice per week, or performed the exercise program in combination with dietary modifications (dietary counseling according to guidelines of the German Nutrition Society (‘healthy diet’) or daily intake of 2 g of *Calanus finmarchicus* oil (CO)).

## 2. Materials and Methods

### 2.1. Participants and Study Design

For this analysis, a subset of participants was randomly selected from participants who took part in a larger single-center, randomized controlled trial in a parallel group design [29]. In the original study, 134 participants between 50–70 years were recruited from the general population in Hannover, Germany, via advertisements in local newspapers and on public notice boards.

Participants had to meet the following inclusion criteria: no exercise training aside from the daily activities for at least two years, a stable body weight (±5 kg) for at least six months, ability to physically perform the exercise intervention (exercise capacity) and consumption of an omnivorous diet. Participants were excluded from the study if they met one of the following exclusion criteria: (suspected) diagnosis of cardiovascular disease (angina pectoris, myocardial infarction, stroke, peripheral arterial occlusive disease, heart failure, cardiac arrhythmia), type 1 and 2 diabetes mellitus, renal insufficiency and liver disease, coagulation disorders, chronic gastrointestinal disorders (e.g., Crohn’s disease), pancreatic insufficiency, immunological disease (e.g., autoimmune disease), intake of immunosuppressive drugs or laxatives, intake of supplements containing *n*-3 PUFAs, alcohol, drug and/or pharmacological abuse, pregnancy or lactation, retraction of informed consent by the subject, concurrent participation in another clinical study or participation in another study in the last 30 days. Eligibility was assessed using a screening questionnaire, while cardiovascular health was assessed by resting and exercise electrocardiography, supervised by trained professionals and a physician.

Ethical approval was granted by the Ethics Commission of the Medical Chamber of Lower Saxony (Hannover, Germany). In accordance with the guidelines of the Declaration of Helsinki, written informed consent was obtained from all participants prior to their participation in the study. This study was registered in the German Clinical Trial Registry (DRKS00014322).

### 2.2. Exercise and Dietary Interventions

In brief, the participants were randomly assigned to one of four study groups by an independent researcher using stratified randomization according to the covariates (sex, BMI, age): (1) control group (CON), (2) exercise only group (EX), (3) exercise and dietary counseling group (EXDC), (4) exercise and CO supplementation group (EXCO).

Exercise training was performed for 12 weeks in fitness centers and consisted of a warm-up and two passes of a strength endurance circuit. The strength endurance circuit consisted of machine-supported strength exercises and bouts of aerobic training. All machines within the circuit required insertion of a chip card which ensured that the machine parameters were adjusted according to the information on the chip card. At the first visit in the fitness center, all participants received an instruction from a trainer and their chip card was programmed. Thereafter, training was conducted independently.

The strength training consisted of six machine-supported exercises that included all major muscle groups and were performed for 1 min each. During the initial training session, a supervised maximum force test with three tries was performed, respectively. The best of the three tries was scored and used to set the machine to 60% of the participants maximum force for the first two weeks of training. For the subsequent six weeks, the load was increased by 10%, and again by 5% for the last four weeks. The endurance exercise consisted of two four-minute bouts per pass, performed on bicycle ergometers and cross-trainers. During the initial session, participants were instructed to aim for perceived exertion that equaled a value of 15 on the Borg scale. Between different exercises, the participants had 30 s of rest. Including the warm-up and rest periods, the training session could be completed in approximately 1 h. Compliance of the participants was assessed via a training log and a questionnaire at the end of the study.

While the EX group performed the exercise program only and was asked to maintain their habitual diet, the EXDC group received dietary counselling in accordance with the guidelines of the German Nutrition Society [30] prior to initiation of the exercise program. Participants from the EXCO were asked to maintain their habitual diet supplemented with capsules providing 2.0 g of oil from *Calanus finmarchicus* (Calanus AS, Tromsø, Norway) per day. The capsules provided 109 mg EPA, 87 mg DHA, and 3.6 mg astaxanthin (a detailed overview of the lipid profile was already reported elsewhere [31]).

Compliance of participants and adherence to the instructions was monitored via fortnightly phone calls. Additionally, participants from the EXCO group were instructed to return leftover capsules. Participants had to consume at least 90% of the capsules to be considered as compliant. In addition, compliance could be additionally verified by an increased omega-3-index detected in the EXCO group [31].

### 2.3. Dietary Intake

The dietary behavior was monitored via 3-day dietary records at the beginning, after six weeks, and at the end of the study after 12 weeks. Nutritionists checked the records for completeness, readability, and plausibility. The software PRODI6.4® (Nutri-Science GmbH, Freiburg, Germany) was used to estimate the energy and nutrient intake. Food groups were analyzed with a food frequency questionnaire (FFQ) from the German Health Examination Survey for Adults (Studie zur Gesundheit Erwachsener in Deutschland, DEGS) of the Robert Koch Institute.

### 2.4. Sample Collection and Preparation

Venous blood samples were drawn from the participants after an overnight fast (≥10 h) at the beginning (pre) and after the intervention (post). For sirtuin analysis, 2 mL blood was collected using EDTA tubes (Sarstedt AG & Co. KG, Nümbrecht, Germany). For RNA isolation, 500 µL of EDTA-blood was transferred into RNAprotect Animal Blood Tubes (Qiagen, Hilden, Germany). The remaining blood was centrifuged at 3300× *g* for 3 min and the plasma used for analysis of sirtuins.

### 2.5. Sirtuin Activity Assay

Deacetylase activity of SIRT1 and SIRT3, as well as desuccinylase activity of SIRT5 in plasma, were assayed with SIRT1, SIRT3 and SIRT5 fluorometric drug discovery assay kits (Enzo Life Science, Lörrach, Germany). The enzyme capacity was measured under substrate saturation which was achieved by addition of a surplus of NAD+ to the assays. Subsequently, plasma samples were diluted 1:5 in HEPES buffer (110 mM NaCl, 2.6 mM KCl, 1.2 mM KH_2_PO_4_, 1.2 mM MgSO_4_x7H_2_O, 1.0 mM CaCl_2_, 25 mM HEPES) according to the manufacturers protocol, and then sonicated for 10 s at 20 kHz with an amplitude of 75% to break cell membranes.

Total protein concentration of the analyzed samples was measured with the Pierce™ BCA Protein Assay Kit (Thermo Fisher Scientific, Waltham, MA, USA), for normalization of the detected SIRT-activity signals.

We are aware that the assays used rely on artificial substrates. Therefore, translation of the results to the in vivo situation is limited.

### 2.6. RNA Isolation and qRT-PCR

RNA was isolated with RNeasy Protect Animal Blood Kit (Qiagen) according to the product protocol. RNA quantity and quality were assessed using a NanoDrop™ 2000 measuring E_260_ and ratio E_260_/E_280_. The isolated RNA was reverse-transcribed to cDNA with the Omniscript RT Kit (Qiagen). Real-time PCR of different cDNA samples was carried out with SYBR green on a QuantStudio™ 6 Pro real-time PCR system (Thermo Fisher Scientific, Waltham, MA, USA). Primers used in this study are shown in Appendix A. Relative changes in the mRNA expression were calculated according to Vandesompele et al. [32], with SUPT20H as a reference for relative quantification.

### 2.7. Data Analysis and Statistical Methods

As already stated before, a subset of participants was randomly chosen for this study from a larger study group. Sample size (*n* = 7 per group) was calculated based on our previous results obtained for sirtuins measured in human blood [14], assuming a two-sided level of significance of 5% and a power of 80%. In total, 40 samples were analyzed.

Baseline data and data of dietary intake are presented as mean ± standard deviation (SD). Sirtuin activity and relative expression are presented as median (+max/−min). Due to the small sample size per group, we performed non-parametric analysis only. Within group differences were analyzed with the Wilcoxon test, differences between sexes using the Mann–Whitney U test and group comparisons using the Kruskal–Wallis test. If significant group differences were detected, a post hoc analysis with Bonferroni correction was performed. Absolute differences (Δ) were calculated as post-pre. To determine correlations between changes in dietary intake and sirtuin activity, Spearman correlations were performed. *p*-values of <0.05 were considered as significant. All statistical analyses were carried out using SPSS software (version 23.0; SPSS Inc., Chicago, IL, USA). Graphs were created with Prism 9 (GraphPad Software, La Jolla, CA, USA).

## 3. Results

### 3.1. Baseline

Baseline characteristics of the participants from the respective subgroups are shown in Table 1. No statistically significant differences regarding distribution of gender, age, height, weight or BMI were found between the study groups.

### 3.2. Dietary Intake

Regarding the changes in dietary intakes estimated from three-day dietary food logs, no significant differences neither within nor between the four study groups were detected (Appendix A). However, a trend to a decreased energy intake was observed in the three intervention groups performing exercise. The highest decrease of approximately 16% was observed in the EXDC group (1955 ± 527 to 1634 ± 528 kcal, *p* = 0.039). In line with those findings, the highest weight loss was observed in the EXDC group (CON: −0.01 ± 1.57, EX: −1.09 ± 1.99, EXDC: −1.54 ± 1.34 kg and EXCO: −0.09 ± 0.7 and respectively, *p* = 0.051).

Further within-group comparisons showed a significant increase in α-tocopherol in CON (6.61 ± 2.03/day to 8.85 ± 2.42 mg/day, *p* = 0.004), as well as a decrease in fiber (24.05 ± 8.18 to 19.32 ± 7.37 g/day, *p* = 0.030) and vitamin C (165.2 ± 102.2 to 106.0 ± 38.0 mg/day, *p* = 0.035) in the EX group.

Analysis of intakes of food groups showed a significant increase in fruit (1.49 ± 1.21 to 2.52 ± 1.26 portion/day, *p* = 0.016) and an increased intake of vegetables (0.87 ± 0.28 to 1.44 ± 0.82 portion/day) in EXDC (Appendix A). Increased intake of vegetables did not reach statistical significance (*p* = 0.070). In the CON group, a significant decrease of vegetable intake (1.80 ± 3.15 to 0.72 ± 0.58 portion/day, *p* = 0.031) and cereal products (3.82 ± 2.41 to 2.44 ± 0.80 portion/day, *p* = 0.039) occurred. Comparison of absolute intakes showed that intake of vegetables differed between the four study groups (*p* = 0.018), and EXDC showed a significantly higher intake in daily vegetable intake than CON (0.57 ± 0.66 vs. −1.08 ± 2.67, *p* = 0.019).

### 3.3. Sirtuin Activity

Sirtuin activities in plasma were analyzed in vitro (under substrate saturation) before and after the 12-week intervention. At baseline, no gender differences and no differences between groups for activities of SIRT1 and SIRT5 were found. For SIRT3, a significant difference among the groups could be detected (*p* = 0.035). However, after Bonferroni correction for multiple comparisons, the differences were no longer significant (CON vs. EX, *p* = 0.101; CON vs. EXDC, *p* = 0.144; CON vs. EXCO, *p* = 0.059). To account for potential effects of baseline differences in the capacity of SIRT1 and particularly SIRT3, correlation analyses of baseline values with absolute differences were performed for each group. No significant correlations were detected in the exercising study groups.

Within group comparisons of sirtuin activities before and after the intervention showed that activities of particularly SIRT1 and SIRT3 increased significantly within the three intervention groups performing exercise, while no significant changes occurred in the CON group (Figure 1A–C). For SIRT1, activity increased in EX from 0.87 U/mg (+1.42/−0.19) to 0.99 U/mg (+1.99/−0.24) (*p* = 0.002), in EXDC from 0.86 U/mg (+1/−0.64) to 1.09 U/mg (+1.52/−0.83) (*p* = 0.008), and in EXCO from 0.96 (+1.13/−0.35) to 1.13 (+1.92/−0.75) U/mg (*p* = 0.012) (Figure 1A). In a similar manner, SIRT3 increased in EX from 2.44 U/mg (+2.96/−1.25) to 3.09 U/mg (+6.13/−1.54) (*p* < 0.001), in EXDC from 2.36 U/mg (+3.01/−1.59) to 3.23 U/mg (+6.53/−2.24) (*p* = 0.008), and in EXCO from 2.52 U/mg (+3.09/−1.84) to 4.02 U/mg (+5.56/−2.93) (*p* = 0.004, Figure 1B). SIRT5 only increased significantly in EX from 0.24 U/mg (+0.51/−0.16) to 0.28 U/mg (+0.63/−0.18) (*p* = 0.001) and in EXCO from 0.32 U/mg (+0.56/−0.12) to 0.32 U/mg (+0.61/−0.16) (*p* = 0.012, Figure 1C).

Comparison of absolute differences in sirtuin capacity revealed significant differences among the study groups for all three sirtuins analyzed (Figure 2). As shown in Figure 2A, SIRT1 activity increased by 0.15 U/mg (+0.56/−[−0.16]) in EX, 0.25 U/mg (+0.52/−0.06) in EXDC and 0.40 U/mg (+0.88/−[−0.12]) in EXCO. Of those, EXCO and EXDC differed significantly from CON (CON vs. EXCO, *p* = 0.003; CON vs. EXDC, *p* = 0.010). Regarding the activity of SIRT3, EX showed an increase by 0.80 U/mg (+3.18/−0.05), EXDC by 0.95 U/mg (+3.88/−0.55) and EXCO by 1.60 U/mg (+2.85/−0.70) (Figure 2B). All increases were significantly different from CON (CON vs. EX, *p* = 0.007; CON vs EXDC, *p* < 0.001, CON vs. EXCO, *p* = 0.004). Activity of SIRT5 increased in all study groups (Figure 2C). EX differed significantly from CON (*p* = 0.032).

### 3.4. Relative Expression

In addition to enzyme activity, relative expression levels of SIRT1, SIRT3 and SIRT5 were analyzed at the beginning and at the end of the intervention. Due to unsuccessful RNA extraction from some samples, only a total of *n* = 6–12 participants per group could be analyzed.

There were no differences in baseline expression levels and no differences between genders. Analysis of the changes in relative expression showed no significant differences among the four study groups (Figure 3).

Although we could not detect any significant differences in dietary intake apart from increased vegetable intake in the EXDC group, we examined the potential impact of different dietary components on sirtuin activity. Therefore, we examined correlations between sirtuin capacity and caloric intake, as well as food groups containing polyphenols or flavonoids (fruits, vegetables, cereals, green and black tea, coffee, wine). This analysis did not reveal any significant correlations. As we could already observe a significant impact of intake of antioxidative substances on short-term exercise-induced sirtuin capacity in recreational runners with different diets [14], we also examined the intake of such substances in this analysis. However, the sirtuin capacity seemed to be unaffected by dietary intake of tocopherol, vitamin C or vitamin A (expressed as retinol equivalent). Additionally, intake of EPA and DHA, metabolic markers of glucose or fatty acid metabolism and anthropometric data did not show any correlations.

## 4. Discussion

To the best of our knowledge, this is the first investigation on the impact of chronic exercise combined with dietary modifications (dietary counselling according to the German nutrition association vs. CO supplementation), on sirtuins measured in human peripheral blood. Sirtuins were already reported to be impacted by chronic exercise, as well as dietary modifications such as caloric restriction or dietary components such as polyphenols [33].

### 4.1. Exercise and Sirtuins

We hypothesized that chronic exercise over 12-weeks would upregulate sirtuin activity and that upregulation would additionally be enhanced by dietary modifications (dietary counselling vs. CO intake). A previous study from our group found differences in short-term exercise-induced sirtuin activity in recreational athletes with different diets (omnivorous, lacto-ovo-vegetarian, vegan) following an exercise test on a bicycle until voluntary exhaustion [14]. In this study, sirtuin activity was also measured in peripheral blood, which gave first evidence that measurement of blood sirtuin levels may possibly reflect sirtuin tissue levels.

Results from the present study suggest that participation in a 12-week exercise program consisting of combined resistance and aerobic training indeed upregulated the activity of SIRT1 and SIRT3 in healthy, elderly, overweight participants as an adaptation to meet increased energy demand. This is also supported by results from Villanova et al., who reported that sirtuin deacetylase activity is higher in athletic individuals when compared to non-trained individuals [34]. Generally, our results are in line with animal and human studies, that showed a positive impact of acute and chronic exercise on sirtuins [13,35,36]. In that context, the most studied sirtuins are SIRT1 and SIRT3 which regulate many metabolic processes that are upregulated during acute exercise, namely energy metabolism, but also mediate systemic long-term adaptations to exercise [37,38,39]. SIRT3 and SIRT1 are involved in the regulation of fatty acid oxidation and mitochondrial biogenesis, while SIRT1 is also involved in the regulation of glucose homeostasis [40,41]. Up-regulation of SIRT5 by exercise was less pronounced and inconsistent. SIRT5 is known to regulate the citric acid cycle, fatty acid oxidation and the mitochondrial respiratory chain [42].

In general, up to now, it is not fully clear to which extent sirtuins in plasma reflect sirtuin function in tissues. However, recent findings from Braud et al. demonstrate that circulating levels of SIRT1 measured in plasma are exclusively linked to visceral adipose tissue in mice [43]. Therefore, it also has to be taken into account that changes in sirtuins in the present study may also be mediated through loss of adipose tissue, as indicated by the amounts of weight lost in the intervention groups. Nonetheless, weight loss was not equal between the three intervention groups (EXDC > EX > EXCO), but EXDC and EXCO seemed to show additive effects of exercise and dietary modifications, despite the little amount of weight loss in EXCO.

Regarding studies investigating the effect of exercise on sirtuins, animal studies analyzed sirtuins in tissues such as the brain, skeletal muscles, heart, liver or adipose tissue [44,45,46]. When measured in human skeletal muscle, two weeks of high intensity training (HIIT) increased the activity of SIRT1, while the protein content stayed the same [47]. This is in line with findings from our study, which also showed an increased activity of SIRT1 in plasma. However, we could not measure sirtuins at protein level as there was a sustained shortage of antibodies. Protein expression levels of SIRT3 in the vastus lateralis of young and older subjects that were classified as sedentary (defined as <30 min exercise a day, not more than twice per week) or trained (defined as 1 h running or cycling, 6x/week for the last four years) showed that the sedentary subjects had lower levels than the trained subjects [48].

Based on those findings, it can be hypothesized that the exercise frequency and intensity used in our intervention was not sufficient to induce changes in gene expression. Moreover, short-term regulation of sirtuin activity are also likely to be driven by post-translational modifications.

When we compared basal enzyme capacities of SRT1, SIRT3 and SIRT5 in the present study with our previous study [14], enzyme capacities were higher in the younger participants (18–35 years) than in the older participants (50–70 years) in the present study. This is in line with findings of Villanova et al. [34] and Lalia et al. [21].

The underlying mechanisms for the exercise-induced increase of the enzyme activity are not fully elucidated, but may be mediated by exercise-induced changes of the NAD+/NADH- or AMP/ATP-ratio [49,50]. This explanation is supported by results from Lamb et al., 2020, who reported that 10 weeks of resistance training (2×/week) resulted in increased muscle NAD+-levels and higher global sirtuin activity in middle-aged, untrained participants [51]. Although those results cannot be compared directly to our results due to the different methodologies in performing exercise and measuring sirtuin activities, the results are coherent.

### 4.2. Diet and Sirtuins

In the present study, diet had an additional impact on sirtuin up-regulation during exercise. The activities of SIRT1 and SIRT3 increased to a higher extent in EXDC and EXCO when compared to EX.

Sirtuins are known to be regulated by polyphenols, which are abundant in vegetables, fruits, whole-grains, coffee, green tea and wine [19]. Indeed, there was a significant increase in fruit and vegetable intake in the EXDC group after dietary counseling. However, in the case of SIRT1, the stability and metabolization of polyphenols are important [52], which has not been examined in our study. Therefore, the efficacy of increased vegetable and fruit intake is difficult to assess without specification of actual intake of polyphenolic compounds. In addition to an increased vegetable and fruit intake, the EXDC group also showed the highest reduction in caloric intake, which is another potential regulator of sirtuins.

In the EXCO group, the intake of PUFAs may be responsible for the increased response of sirtuin activities to exercise. Interestingly, supplementation of *n*-3 PUFAs has been reported to positively impact muscle strength and function in older adults [53,54]. Sixteen weeks of 3.6 g/day *n*-3 PUFA intake (2.7 g EPA + 1.2 g DHA) increased myofibrillar and mitochondrial protein synthesis measured after a single bout of resistance exercise in older adults [21], which supports the idea that exercise combined with *n*-3 PUFA supplementation may also enhance long-term adaptations mediated by sirtuins. Generally, *n*-3 PUFAs are not only discussed to support muscle anabolism [55], but also modulate processes such as adaptive thermogenesis through activation of the G-protein coupled receptor 120 [56]. Among others, SIRT3 is one of the markers shown to be upregulated in response to GPR120 activation [57]. However, it has to be acknowledged that CO used in the present study provided only approximately 200 mg of EPA + DHA, which is far below the doses which reportedly have beneficial effects. It is already known that *n*-3 PUFAs can modulate metabolic pathways via activation of transcription factors, namely the peroxisome proliferator-activated receptors (PPARs) [58]. This process may be mediated by sirtuins. Indeed the *n*-3 PUFAs, EPA and DHA, were linked to the activation of SIRT1- and SIRT3-dependent pathways [24,25,26,27]. Another fatty acid, which was reported to impact SIRT1-associated pathways is the monounsaturated omega-9 fatty acid oleic acid [59], which is also present in CO (36 mg oleic acid per 2 g of CO). Lastly, CO contains the antioxidant astaxanthin. There is evidence from animal studies indicating that astaxanthin increases the expression of SIRT1 [60] and the mitochondrial sirtuins [61]. Both, the antioxidative properties and the potential impact of *n*-3 PUFAs on sirtuins may be responsible for the beneficial effects of CO on obesity-related disorders like insulin-resistance and atherosclerosis, as previously reported in animal studies [62,63].

As increased body weight and metabolic disorders such as type 2 diabetes were already shown to correlate with sirtuin blood levels [64,65], we also investigated whether there were any links between body weight, BMI and metabolic markers using correlation analysis. As the study collective was very homogenous and metabolically healthy, it seems not surprising that no significant correlations were detected. To examine the potential role of blood sirtuins as markers for health, it would be interesting to perform analyses in different age groups and in patients with and without metabolic disorders. Although it has to be considered that blood sirtuins do not directly reflect tissue levels, establishment of reference ranges in different study populations would be helpful to assess blood sirtuins as biomarkers.

Overall, this study has limitations. As mentioned above, we are aware that plasma sirtuin activities do not necessarily reflect tissue activities of sirtuins. However, to investigate the effect of different tissues on sirtuins, mRNA needs to be measured in secreting tissues which requires the application of biopsies. As biopsies are invasive, they are not always applicable and ethically justified. Furthermore, it is important to point out that the Fluor de Lys assay which was used in this study relies on an artificial substrate, thus may not reflect enzyme activities in vivo [66]. Therefore, the results presented on sirtuin activities need to be interpreted with caution and further studies on the interrelation between body tissues, plasma levels of sirtuins and the impact of diet and exercise on both components are needed. Lastly, the evaluation of the impact of dietary modifications on sirtuin activity would have been best under strictly controlled dietary conditions (e.g., meal plans). However, the use of ad libitum diets (including dietary recommendations for the EXDC group) make the results obtained more transferable to real-life scenarios. Ultimately, future research needs to elucidate whether or not plasma sirtuins reflect tissue levels.

## 5. Conclusions

In summary, our study showed that 12 weeks of twice-weekly combined resistance and aerobic training up-regulated the capacities of SIRT1 and SIRT3 in untrained, overweight, elderly subjects; while the capacity of SIRT5 was less affected. The dietary modifications seemed to further enhance the effect of exercise-induced sirtuin activation. However, further research on plasma and tissue sirtuins is needed to evaluate the validity and applicability.

## Figures and Tables

**Figure 1 nutrients-13-03824-f001:**
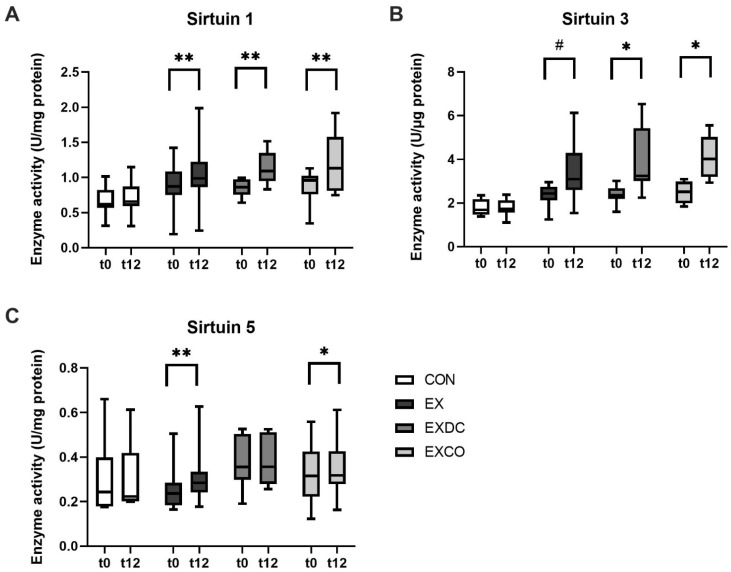
Absolute enzyme capacity (under substrate saturation) of SIRT1, SIRT3 and SIRT5. Enzyme activity was measured before (pre) and after (post) the intervention in the four study groups, CON, EX, EXDC and EXCO, for SIRT1 (**A**), SIRT3 (**B**) and SIRT5 (**C**). Data are shown as median ± quartiles and extremes; *n* = 8–14. To assess within group differences a Wilcoxon test was conducted (* *p* < 0.05; ** *p* < 0.01; # *p* < 0.001).

**Figure 2 nutrients-13-03824-f002:**
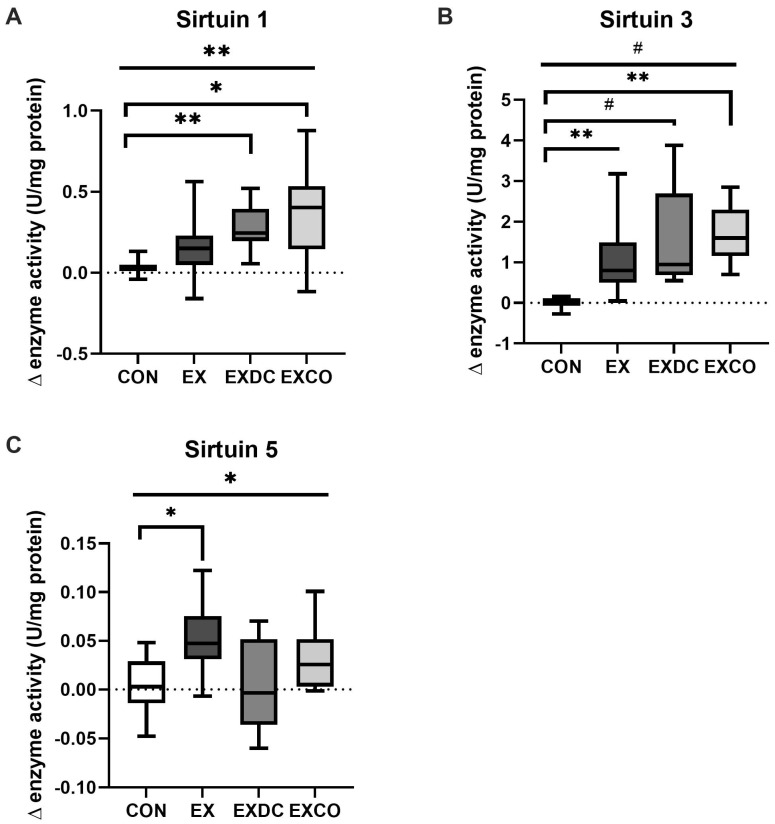
Changes of enzyme capacity (under substrate saturation) of SIRT1, SIRT3 and SIRT5. Changes were calculated as the difference of enzyme capacities after (post) and before (pre) the intervention for the four study groups, CON, EX, EXDC and EXCO, for SIRT1 (**A**), SIRT3 (**B**) and SIRT5 (**C**). Data are shown as median ± quartiles and extremes; *n* = 8–14. To assess statistical differences a Kruskal–Wallis test was conducted and if significant group differences were detected, a post hoc analysis with Bonferroni correction was performed (* *p* < 0.05; ** *p* < 0.01; # *p* < 0.001).

**Figure 3 nutrients-13-03824-f003:**
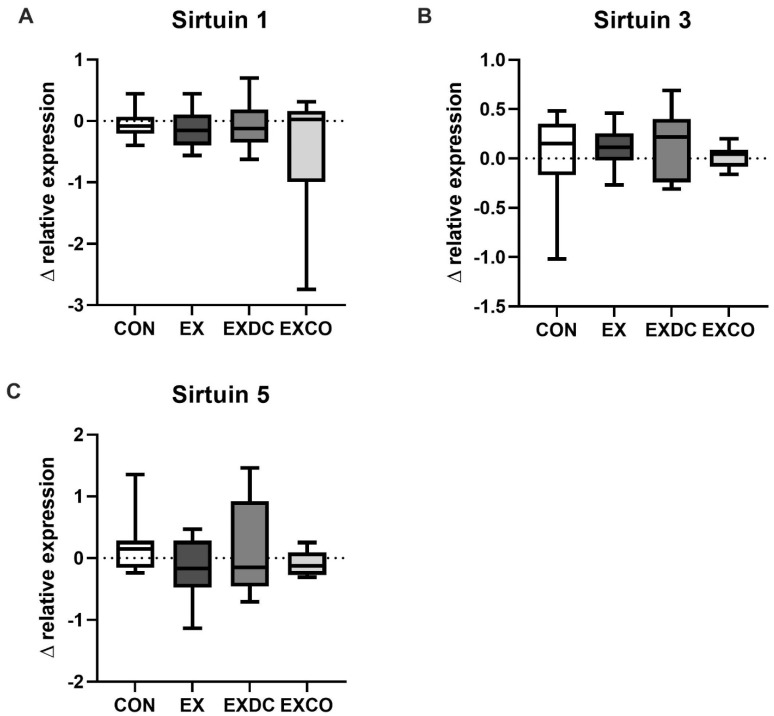
Changes in the relative expression of SIRT1, SIRT3 and SIRT5. Changes were calculated as the difference of relative expression after (post) and before (pre) the intervention for the four study groups, CON, EX, EXDC and EXCO, for SIRT1 (**A**), SIRT3 (**B**) and SIRT5 (**C**). Data are shown as median ± quartiles and extremes; *n* = 6–12. To assess statistical differences a Kruskal–Wallis test was conducted.

**Table 1 nutrients-13-03824-t001:** Baseline characteristics of all participants.

	CON (*n* = 9)	EX (*n* = 14)	EXDC (*n* = 8)	EXCO (*n* = 9)	*p*
Sex (f/m)	7/2	11/3	5/3	7/2	0.836
Age (years)	61 ± 5	60 ± 6	59 ± 5	60 ± 3	0.825
Height (cm)	166 ± 6	168 ± 7	170 ± 8	174 ± 6	0.112
Body weight (kg)	75.1 ± 13.6	78.4 ± 19.1	86.9 ± 20.0	82.3 ± 16.8	0.503
BMI (kg/m^2^)	27.2 ± 4.1	27.7 ± 6.0	30.0 ± 5.7	27.4 ± 6.0	0.701

Data are shown as mean ± SD. Gender distribution between groups was analyzed using chi-square test. Differences among groups were assessed with Kruskal–Wallis Test. f = females, m = male, BMI = Body Mass Index.

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
