# Peer review of "Impact of Dietary Modifications on Plasma Sirtuins 1, 3 and 5 in Older Overweight Individuals Undergoing 12-Weeks of Circuit Training"

_nutrients, 2021, doi:10.3390/nu13113824_

Round 1
Reviewer 1 Report
Very well written. No major revisions necessary.
This paper is focused on impact of 12-weeks of different physical exercise methods, combined
with dietary modifications on Sirtuins activity in particular in older individuals. Is well structured and easy to read.
A few minor revisions are:
- It would be useful to add a new references about a link between physical activity and
Sirtuin 1 and 3 activation. For better support your data, the following reference should be recommended: “Proposed Tandem Effect of Physical Activity and Sirtuin 1 and 3 Activation in Regulating Glucose Homeostasis” (PMID 31557786).
- Explain why the study is focused only on Sirtuins 1, 3 and 5. Please clarify.
- Check the p-values and the number of matched asterisks.
- It would be appropriate analyze the SIRT’s protein expression in the nucleus, cytoplasm and mitochondria and not only the gene expression.
Reviewer 2 Report
TITLE: Impact of 12-weeks of resistance and aerobic training combined with COMMENTS TO THE AUTHORS:
Wasserfurth P et al examined the effect of 12 weeks of combined aerobic and resistance training with or without dietary counseling or supplementation with Calanus finmarchicus oil on plasma sirtuin activity and blood mRNA expression. The authors report that each of the interventions improved the maximal activity of SIRT1, SIRT3 and SIRT5 in plasma whereas the control group saw no effect. There was no effect of any of the interventions or the CON on blood sirtuin mRNA. While these results are interesting, there are several major issues that preclude me from recommending acceptance of this manuscript for publication in Nutrients. My comments, itemized by section of the manuscript, are listed below. My main concerns are the lack of depth of data leading to uncertainty of the relevance and applicability of these data. For example, the role of plasma sirtuin activity is not well understood in general and this manuscript has not included any data to explore the potential relationships between changes they observe in plasma Sirtuins and phenotypic responses with any of the interventions in this study. Not to mention that several issues have been reported with the Flour de Lys activity assay used in this manuscript for measuring sirtuin activity (10.1111/j.1747-0285.2009.00901.x), citing a risk of the assay being only reflective of sirtuins ability to deacetylate the Fluor de Lys peptide and not be reflective of sirtuins’ activity toward deacetylating targets found in vivo. In addition, the blood measures of sirtuin RNA in the opinion of this reviewer is not additive as it is unlikely that changes in sirtuin RNA in blood cells is unlikely to reflect changes in peripheral tissues such as skeletal muscle, liver or adipose with exercise or dietary intervention. In addition, recent work has demonstrated that circulating sirt1 is primarily, if not exclusively, secreted by adipose tissue (doi: 10.1016/j.redox.2020.101805.) making a measure of sirtuin mRNA in blood cells unfounded.
TITLE:
- The title could be more descriptive.
- Please indicate in the title that sirtuins are being measured in plasma.
- Also is appropriate to describe the demographic as older individuals with overweight or obesity.
ABSTRACT:
- There is no data presented in the abstract.
- There is no conclusion or interpretation of the findings in the abstract.
METHODS:
- It is mentioned that the participants analyzed in this manuscript is part of a subset of a larger trial. Can you please provide a reference to the publication of the primary manuscript if it exists?
- It is unclear if the exercise session were supervised. If so, please provide more clarity on how that was organized. If not, please provide details on how participants were trained on use of machines, self-monitoring of intensity via RPE and self-administration of “maximum force” assessments.
- It is unclear how exactly the resistance exercise was prescribed.
- How many repetitions did the participants perform?
- Were all six machine exercises the same for each participant?
- It is mentioned that all muscle groups were included. What specific exercises were prescribed?
- For the endurance exercise it is unclear if only one 4-minute bout of exercise was performed per session.
- Also, for endurance training, it seems difficult to expect participants to be able to accurately elicit a specific RPE. Is there any heart rate data that the participants were instructed to take that could validate this intensity was reliably reached?
- Were there any pre and post measures of fitness taken (i.e. resting heart rate, submax or max aerobic capacity, strength etc)? If so, please report. Such data are important to demonstrate efficacy of intervention.
- What was the goal of the dietary counseling? Weightloss? Improved food quality? Preservation of muscle mass? All of the above? Participant specific goals?
- As previously mentioned, there are concerns regarding the relevance of the Fluor de Lys assay for reflecting in vivo sirtuin activity. Please acknowledge and address these concerns in the limitations section of the discussion.
- Please provide rationale for measuring blood mRNA of sirtuins. Isther reason to believe sirtuin transcripts in this tissue is relevant to exercise or any clinical outcomes that are meaningful in the context of older individuals with overweight/obesity? In addition, I understand that measuring sirtuin proteins via westerblot was not possible due to limitations in antibody availability but why not use commercially available ELISAs to measure plasma sirtuins? This would improve the depth of data and may be more reflective of peripheral tissue sirtuin especially since recent work in mice suggests adipose as the main source of circulating sirtuins suggesting that blood cell sirtuin transcripts may not be as relevant. This could also explain the lack of correlations that were able to be found between blood mRNA and anthropometric outcomes.
RESULTS AND FIGURES:
- It is mentioned in the results and demonstrated in the supplementary table that the EXDC group reduced their caloric intake by ~16%. Did this correspond with weight loss? This is extremely relevant for the interpretation of the data as caloric restriction is well described to have strong abilities to activate sirtuins especially SIRT1.
- In relation to changes in body weight, are there any other pre – post measures that were taken that can be reported out and attempt to correlate with changes in sirt1 activity? This would also improve the depth of data while also testing the relevance of plasma sirtuin activity with exercise-related changes in important athropometric or fitness outcomes.
- Can the activity data be expressed in U/mg protein so that the numbers are more reader friendly?
- Why is figure 1 not analyzed with a nonparametric 2way ANOVA (Friedman’s test) as in figure 2. This analysis with a Wilcoxon sign-rank test is essentially 4 separate non-parametric t-tests which is not appropriately rigorous.
DISCUSSION:
- The discussion would be better served to not only compare the results of this paper to previous reportings of changes in tissue sirtuins in response to exercise but also present and discuss how these data compare to previous investigations of plasma sirtuins. These investigations are limited and as such it is not clear what if any relevance circulating sirtuins have in response to exercise, diet, in the context of aging or in the context of overweight/obesity. Please expand on the existing literature in this are and how these data add to the current consensus of data.
- Correlations between sirtuin measurments and anthropometric data are mentioned in the discussion. Please add these data in a table as supplementary data.
Round 2
Reviewer 2 Report
Thank you for addressing majority of the critiques provided. My remaining concerns are as follows:
I understand that the difficulties of the pandemic made it impossible to procure ELISAs or antibodies for protein determination. Is it still the case that these reagents are unavailable? If not, are there remaining samples to run an ELISA for any of the sirtuins assessed in this study?
If it is indeed the case that this is the first work to examine sirtuin activity in plasma in response to exercise or otherwise, this should indeed be emphasized in the discussion. However, the statement in lines 338 - 340 is confusing considering your previous study had similar outcomes albeit in a different context. Please clarify this statement so that the novelty of these data are made more apparent.
Author Response
Dear Editors and Reviewer,
we thank you for once more reading through our manuscript and acknowledging the changes made. Following the Editors advice, we carefully went through the manuscript to check for repetitiveness and underwent some language editing. All previous changes within the manuscript are still highlighted in yellow, follow-up changes are made with the “track changes” function.
Regarding the follow-up remarks of Reviewer 2, we address those below.
Reviewer 2:
I understand that the difficulties of the pandemic made it impossible to procure ELISAs or antibodies for protein determination. Is it still the case that these reagents are unavailable? If not, are there remaining samples to run an ELISA for any of the sirtuins assessed in this study?
Unfortunately, the situation got worse during the ongoing pandemics situation, companies expect that shortage of chemicals will continue well into 2022. We appreciate that it would be interesting to do the measurements, however, this will not be possible in the next months.
If it is indeed the case that this is the first work to examine sirtuin activity in plasma in response to exercise or otherwise, this should indeed be emphasized in the discussion. However, the statement in lines 338 - 340 is confusing considering your previous study had similar outcomes albeit in a different context. Please clarify this statement so that the novelty of these data are made more apparent.
We agree that it was not clear to which study we were referencing. We changed the wording to point out the differences between the previous and the present study more precisely (l. 374 and following). Please note, that we also deleted a whole paragraph in the beginning of this part of the discussion (due to repetitiveness). We believe that the discussion is easier to follow in the current form.